# Clam Size Explains Some Variability in Paralytic Shellfish Toxin Concentrations in Butter Clams (*Saxidomus gigantea*) in Southeast Alaska

**DOI:** 10.3390/toxins16110464

**Published:** 2024-10-29

**Authors:** John R. Harley, Kellie Blair, Shannon M. Cellan, Kari Lanphier, Lindsey Pierce, Cer Scott, Chris Whitehead, Matthew O. Gribble

**Affiliations:** 1Alaska Coastal Rainforest Center & Program on the Environment, University of Alaska Southeast, Juneau, AK 99801, USA; 2Environmental Research Lab, Sitka Tribe of Alaska, Sitka, AK 99835, USA; 3Central Council of the Tlingit and Haida Indian Tribes of Alaska, Juneau, AK 99801, USA; 4Department of Medicine, Division of Occupational, Environmental, and Climate Medicine, University of California San Francisco, San Francisco, CA 94143, USA; matt.gribble@ucsf.edu

**Keywords:** harmful algal blooms, paralytic shellfish toxins, PSP, butter clams, Alaska

## Abstract

Harmful algal blooms (HABs) are a reoccurring threat to subsistence and recreational shellfish harvest in Southeast Alaska. Recent Tribally led monitoring programs have enhanced understanding of the environmental drivers and toxicokinetics of shellfish toxins in the region; however, there is considerable variability in shellfish toxins in some species, which cannot be easily explained by seasonal bloom dynamics. Persistent concentrations of paralytic shellfish toxins (PSTs) in homogenized butter clam samples (n > 6, *Saxidomus gigantea*) have been observed in several communities, and relatively large spikes in concentrations are sometimes seen without *Alexandrium* observations or increased toxin concentrations in other species. In order to investigate potential sources of variability in PST concentrations from this subsistence species, we assessed individual concentrations of PSTs across a size gradient of butter clams during a period of relatively stable PST concentrations. We found that increasing concentrations of PSTs were significantly associated with larger clams using a log-linear model. We then simulated six clams randomly sampled from three size distributions, and we determined large clams had an outsized probability of contributing a significant proportion of the total toxicity in a six-clam homogenized sample. While our results were obtained during a period of low HAB activity and cannot be extrapolated to periods of intoxication or rapid detoxification, they have significant ramifications for both monitoring programs as well as subsistence and recreational harvesters.

## 1. Introduction

Butter clams (*Saxidomus gigantea*) are an important subsistence and recreationally harvested bivalve along the northwest coast of North America [1,2,3,4]. Butter clams have been harvested since time immemorial and were actively cultivated via ancient aquaculture practices such as clam gardens by Indigenous groups in the region [2,3,5]. Currently, butter clams are the most frequently harvested and shared marine bivalve species among subsistence harvesters in Alaska, although subsistence use appears to be declining with users citing concerns over harmful algal blooms (HABs) [6,7].

Paralytic shellfish toxins (PSTs), which can cause paralytic shellfish poisoning (PSP), have long been associated with butter clams; the most common PST congener, saxitoxin (STX), is named after the butter clam genus from which it was first isolated [8]. PSTs are produced by a number of dinoflagellate and cyanobacteria species, notably the dinoflagellate *Alexandrium* spp., which is endemic to Southeast Alaska, likely *Alexandrium catenella* [9,10]. There are more than fifty chemical congeners collectively named PSTs that have similar mechanisms of action, but varying toxicities (measured as toxic equivalency compared to STX, henceforth STX-eq) and toxicokinetics [11,12].

Cases of PSP in Alaska are likely underreported due to the lack of healthcare resources in many communities and non-specificity of mild symptoms (e.g., tingling of lips) [13]. Butter clams have been responsible for more reported cases of PSP in Alaska than any other shellfish species [14,15], likely in part due to their high subsistence use rate as well as their propensity to retain PSTs far longer than other bivalves [16,17]. Butter clams have been shown to retain PSTs at concentrations above regulatory thresholds (80 µg STX-eq 100 g^−1^) for several years following initial intoxication, while other species such as blue mussels (*Mytilus edulis* species complex) or cockles (*Clinocardium nuttallii*) will generally fall below regulatory levels within weeks or months following intoxication [7,17,18].

In the absence of a state-supported shellfish toxin monitoring program, several Alaska Native Tribes formed the Southeast Alaska Tribal Ocean Research (SEATOR) network in 2015 to monitor HABs and enhance safety of shellfish harvest (www.seator.org, accessed on 3 August 2024). Participating Tribes collect shellfish samples regularly, targeting 6–15 individuals of each species, and ship samples to the Sitka Tribe of Alaska Environment Research Lab (STAERL) where they are analyzed for PSTs [7]. The individual species are homogenized to achieve 100 g of tissue, which is comparable to the Washington State Department of Health protocols used to sample shellfish, although the DOH uses the mouse bioassay for toxin analysis [19,20]. To date, STAERL has analyzed more than 950 samples of butter clams collected from 18 coastal Alaskan communities, with nearly two-thirds of samples testing over the US Food and Drug Administration (US FDA) threshold of 80 µg STX-eq 100 g^−1^ and nearly 7% testing over an estimated lethal dose for a 60 kg human (0.6 mg, [21]). This represents a significant risk to harvesters and Alaska Natives who are more than 10 times more likely to develop PSP [13,14].

In addition to demonstrating retention of PSTs following HAB events, butter clam toxin concentrations in Southeast Alaska often display significant temporal variability that is inadequately explained by seasonal HAB dynamics inferred from both phytoplankton observations and toxin dynamics in sentinel species (i.e., blue mussels, [18]). For instance, butter clams collected in Juneau, Alaska (58.22° N 134.43° W) by the Central Council of the Tlingit and Haida Indian Tribes of Alaska (CCTHITA) show toxicologically relevant variability in PST concentrations during both winter periods where concentrations in blue mussels are extremely low (often below detection limit, BDL) as well as during depuration periods following significant bloom events (Figure 1). The large blooms observed in 2019 resulted in extremely high toxin concentrations in Juneau clams and were associated with seabird die-offs in nearby colonies [22].

Variability of HAB toxin sequestration and biotransformation within individual bivalves has been described, although the drivers of that variability (genetics, sensitivity to toxin, feeding rate variability, etc.) are not well-understood even in controlled experiments [23,24,25]. Álvarez et al. [26] found significant variability in PSTs in surf clams during a large *Alexandrium* bloom in Chile and suggested sensitivity to PSTs resulting in variable feeding rates could explain inter-individual variability. Two instances of variability in toxin concentrations in butter clams are shown in Figure 1; a more than 2-fold increase in toxin concentrations from 155 to 431 µg STX-eq 100 g^−1^ during December 2018–February 2019 and unexpectedly low concentrations (353 µg STX-eq 100 g^−1^) immediately following a significant bloom in July 2019 with preceding and subsequent samples resulting in concentrations greater than 1000 µg STX-eq 100 g^−1^.

Traditional ecological knowledge (TEK) from Indigenous Salish peoples describes how the siphon and black siphon tip of the butter clam sequesters pollutants and toxins [27], and some studies have found higher concentrations of PSTs in siphon tissues [27,28]. However, this heterogenous distribution of PSTs does not appear to be consistent across individuals (3–39% of total toxin was in the siphon [29]), and the mass of tissue in the siphon is low compared to the overall clam. Thus, this tissular distribution likely does not explain all of the variability seen in regular monitoring programs [27].

Here, we tested one hypothesis that variability in concentrations of PSTs in individual clams can be explained by morphometrics (mass, width) in butter clams collected in Juneau, Alaska, and examined the wider implications of these results for monitoring programs using bootstrapped sampling simulations.

## 2. Results

The median PST concentration from butter clam samples was 83 µg STX-eq 100 g^−1^, just above the US FDA regulatory threshold, with roughly half (46%) of the samples testing below the regulatory threshold (Figure 2). PST concentrations were non-normally distributed, so natural log-transformed concentrations were used in further analyses. No differences in PST concentrations were found over the temporal scope of the study period when data were grouped into sampling efforts (Figure 2b, ANOVA, *p* > 0.01). PST concentrations were generally low during the study period in both blue mussels (*M. trossulus*, collected by CCTHITA as part of the SEATOR monitoring program) and butter clams.

Widths of clam shells ranged from 34 to 108 mm (median 72 mm), and the mass of shucked tissue ranged from 3.87 to 110.02 g (median 35.72 g, wet weight). Shell widths were normally distributed (Shapiro–Wilk test) while mass was not; however, log-transformed mass values were normally distributed. Mass and width were well correlated and a large amount of variance in mass was explained by a log–log relationship with width (*R*^2^ = 0.94, Figure 3a).

Log-transformed PST concentrations were correlated with both mass and width using a linear regression (*p* < 0.001), with the width model explaining slightly more variability in PST concentrations (*R*^2^ = 0.37 for width model, *R*^2^ = 0.33 for mass model, Figure 4b).

The results of bootstrapped sampling simulations are presented in Figure 4. Increasing probabilities of a high total toxin proportion attributed to a single clam were observed with increasing clam masses and widths, and this relationship was strongest in the right-skewed size–frequency distribution (red points) and lowest in the left-skewed size–frequency distribution (blue points).

## 3. Discussion

Butter clams are the most actively harvested and shared subsistence bivalve species in Southeast Alaska, yet their usage appears to be declining due to concerns over HABs and the risk of PSP [6,7]. The SEATOR program has actively monitored concentrations of PSTs in Southeast Alaskan communities since 2016, having analyzed over 3000 shellfish samples and over 950 butter clam samples to date. In that time span, there has not been a case of PSP reported from any of the monitored sites, despite extremely high concentrations of PSTs observed in some years (>4000 µg STX-eq 100 g^−1^) [7,22].

While the lack of PSP incidents is a laudable achievement for SEATOR and the participating Tribes, the concomitant goal of facilitating access to local foods and increasing food security precipitates questions about when and where these resources will be safe to consume. Persistently high concentrations of PSTs in butter clams in some communities and inconsistent trends in toxin dynamics have led to community concerns about the future of access to this important subsistence resource ([30], Figure 1). Thus, seeking to better understand the toxicodynamics of PSTs in subsistence species as well as the potential biases introduced by monitoring protocols is an important priority.

In contrast to prior summers, the sampling period during the summer of 2022 did not have significant concentrations of PSTs in any species monitored by SEATOR nor significant HABs of *Alexandrium* reported in participating communities. While concentrations of PSTs in butter clams may have been slightly decreasing (detoxification) over the course of this study, we did not find statistically significant differences in log-transformed concentrations of PSTs between sampling efforts (Figure 2b). Thus, in the absence of significant *Alexandrium* toxin production, and in the absence of significant detoxification, the sampling period assessed here can be characterized as similar to the generally stable concentration dynamics observed in routinely monitored butter clams from this area.

Our results suggest that, under the conditions observed in 2022, butter clam PST concentrations were significantly positively correlated with butter clam size (Figure 3b). While the log-linear model fit does not perfectly explain variability in toxin concentrations observed (*R*^2^ = 0.37), the relationship is statistically significant (*p* < 0.01). In the upper quartile of clam width, 81% of samples (13 of 16) had concentrations above the regulatory threshold of 80 µg STX-eq 100 g^−1^, while in the lower quartile, only 19% (3 of 16) had concentrations above that threshold. Median concentrations of PSTs (83 µg STX-eq 100 g^−1^) were low in this study compared to median concentrations in butter clams from Juneau since 2016 (230 µg STX-eq 100 g^−1^, SEATOR data). This is likely partially explained by the absence of a large bloom during the sampling period (Figure 2a), but is also partially explained by the differences in sampling between this study and regular monitoring conducted by the CCTHIA. In the present study, we purposefully selected butter clams across a size gradient. While SEATOR sampling protocols do not directly address size, during regular monitoring, clams are often sampled with a preference towards larger individuals, which more closely reflects harvesting practices and reduces the impact on the sampling site by reducing the necessary number of clams to reach the requisite mass for the assay (100 g).

There have not been many studies to assess morphometrics in relation to total toxicity in individual homogenized bivalves. While some studies have assessed the distribution of toxins within individuals, consistent patterns in tissular distribution have not always been observed [28,29]. Several studies have found that the toxin concentration and size are inversely related; for instance, smaller mussels were found to have higher concentrations of PSTs [31,32]. The explanation for this effect, that the ingested phytoplankton/toxin mass represents proportionally more of the total mass of the bivalve in smaller individuals, would also apply to butter clams. However, there are key differences that can explain the opposite pattern observed in our study. The toxicokinetics of PSTs in butter clams are markedly different than in mussels; where mussels show rapid elimination/biotransformation of PSTs, butter clams are well known to retain appreciable concentrations of PSTs for several years following intoxication [17]. No pattern was discerned between size and toxicity of butter clams in an early study from British Columbia [33], although that study had a low statistical power to detect differences in toxin concentrations.

The receptor binding assay (RBA) used in this analysis does not provide congener-specific concentrations, and it is possible that some of the differences in toxicity as measured by the RBA could be due to differences between binding affinities of PSP congeners in the RBA assay and observed toxicity in animal models. Detoxification and elimination of PSTs are often congener, species, and tissue specific [34]. However, we feel that there are several reasons to suspect that larger clams having higher proportions of high-binding-affinity, low-toxicity congeners is likely not the sole explanation for the observed relationship between toxicity and size.

Firstly, in butter clam samples from Alaska that have been analyzed via high-performance liquid chromatography (HPLC), STX is often the dominant congener in terms of molar fraction [35,36]. Kibler et al. found a higher proportion of total toxicity was contributed by gonyautoxins (GTXs) in summer months, attributed to active blooms of *Alexandrium* (and accompanying increases in total toxicity), which were not observed in our study [29]. In that study, for most of the year, the majority of total toxin concentration was attributed to STX. Butter clams are able to retain STX for long periods of time due at least in part to resistance to the toxic effects of PSTs [16], and it has been suggested that this insensitivity may explain the lack of biotransformation to less toxic PST congeners in butter clams compared to other species [31,37].

The binding affinities and toxic equivalence factors of PSP congeners are not well-resolved and are highly dependent on myriad factors such as routes of exposure and assay-specific factors such as receptor variability and different animal models used for membrane homogenates [38,39,40]. However, some studies have found that the binding affinity for major congeners is similarly ordered to toxicity, as determined in the mouse model e.g., [8], and the total toxicity determined via the HPLC method does show a good correlation with the RBA for the determination of the total toxicity in STX-equivalents in this and other bivalve species [35,36,40,41]. Although there may be discrepancies between binding affinities and observed toxicities for some congeners, the agreement of RBA and HPLC across a wide range of total toxin concentrations suggests that results obtained from the RBA are not severely overestimating total toxicity as compared to HPLC. This supports findings from several studies [35,42] that found low-toxicity PSP congeners (e.g., GTX5) are not found in appreciable concentrations in this species. Thus, while the idea that variation in toxicity in individual clams may be related to congener-specific profiles, we do not think this is the mechanism driving a significant amount of the variability we see across a size gradient here. However, further research into size-specific PST congener profiles and biotransformation is certainly warranted.

The range of morphometrics and relationship between mass and shell size described here is similar to modeled relationships for butter clams from Puget Sound [43]. In order to investigate the impact of the size–PST relationship in butter clams on observed variability in toxin concentrations from the SEATOR monitoring program, we examined three hypothetical sampling scenarios. We used three beta-distributions to simulate sampling methods—a right-skewed distribution (red, Figure 4), which is reflective of a sampling scheme where smaller individuals are targeted, either to favor larger clams for consumption or reflective of a natural right-skew in size distributions, which is observed in many species [44]. A left-skewed distribution (blue, Figure 4) simulating preferentially selecting large clams for sampling, which may reflect a higher encounter rate while digging or a desire to accumulate sufficient sample mass required for toxin testing. And finally, a symmetric distribution (yellow, Figure 4) reflective of either a random sampling from a normally distributed population or a targeted sampling scheme to reflect a wide size distribution, as in this study. In each modeled distribution, larger clams were more likely to contribute a disproportionate amount of toxin to the homogenized sample of six clams. This effect is more dramatic in the right-skewed distribution (red), where a single clam has the potential to contribute >50% of the total toxicity in a homogenized sample of mostly smaller individual clams. This effect is lessened in the left-skewed distribution (blue), where a sample of mostly larger clams will have a more even distribution of toxin contribution from each individual clam.

We emphasize that while we found a significant relationship between clam size and toxin concentration in butter clams in this study, this study occurred during a period of relatively little HAB activity in the region, and toxin concentrations were generally low compared to their long-term average. It is unknown if this relationship between size and toxin concentration would hold in butter clams during periods of intoxication (active bloom) or rapid detoxification immediately following a bloom. Variable detoxification rates have been seen in some species in field settings, although the drivers of the variation (environmental factors, genetics, food availability) have not been well-described [45]. While these questions are an interesting area of future research, we emphasize that with TEK, HAB monitoring programs, and communication and outreach efforts, subsistence and recreational shellfish harvest is more common outside of these traditional bloom windows [4,7,46]. Therefore, the variability of shellfish toxins in butter clams during periods of relatively stable toxin concentrations (e.g., winter) is also important for both monitoring efforts and continuing safe subsistence harvest. Even though further study into small-scale dynamics of toxin accumulation in shellfish is warranted, it is also important for monitoring programs such as SEATOR to regularly evaluate their monitoring methods to ensure toxin data and advisories produced are providing relevant information to harvesters.

## 4. Conclusions

We found evidence that the PSP toxin concentration was positively correlated with butter clam size, and that large butter clams can contribute a disproportionate amount of toxin to homogenate samples collected for routine monitoring. If size–toxin relationships exist under other conditions and in other regions, these results could have implications for harvesters and HAB monitoring programs.

## 5. Materials and Methods

Butter clams were collected during negative tides (low tidal height less than 0 m) in five sampling efforts from 15 June to 15 August 2022. Clams were collected from three locations near Juneau, Alaska approximately 5–10 km apart under Alaska Department of Fish and Game (ADFG) permit CF-23-069 and opportunistically as part of recreational harvest. Clams were dug using rakes and by hand in rocky intertidal areas, with 6–12 individuals taken during each sampling effort, totaling 70 clams across the study period.

Clams were not randomly selected; rather, after extracting at least 10 clams from each beach, individuals were selected to represent a size gradient. Clams were placed in a cooler and transported back to the lab, where they were measured for width across the widest axis of their shells using calipers to the nearest mm. All soft tissues were then removed and allowed to drain across a fine wire sieve, and soft tissues were subsequently weighed and placed in Whirl-Paks. Clams were then placed in a −20 °C freezer. Following the end of the field season, samples were removed from the freezer, briefly allowed to thaw, and homogenized using an immersion blender or microhomogenizer before refreezing. Samples were then shipped on ice to STAERL for PST analysis.

Samples were analyzed for PSTs using a receptor binding assay (RBA), which uses a competitive binding technique to assess the toxin content of shellfish samples [36,47]. The procedure used by STAERL is described more thoroughly in [7]. Briefly, samples are added to microplates prepared with porcine brain membrane homogenate and radiolabeled saxitoxin ([H3] STX, American Radiolabeled Chemicals, St. Louis, MO, USA), and a microplate counter is used to measure bound [H3] STX against a standard curve. Samples were run in triplicate, and arithmetic means of replicates were used in further analyses. The coefficient of variation averaged 9% across triplicates. At least two quality control standards run in triplicate were included in each plate, and the percent recovery of standards averaged 118%. The limit of quantification averaged 3.9 µg 100 g^−1^ STX-eq across all plates.

Concentrations of toxins were analyzed in association with morphometrics (shell width and shucked mass). One sample was BDL, and a substituted concentration of LOQ*√2 was used in further analysis. A log-linear regression was fit to both PST-mass and PST-width models. To investigate the influence of individual clam size on homogenized sampling methods such as those employed by SEATOR, we ran bootstrapped simulations using three hypothesized size distributions of butter clams. Three beta distributions were used for sampling simulations: one approximating a Gaussian distribution (α = 3, β = 3), a right-skewed distribution (α = 3, β = 9), and a left-skewed distribution (α = 9, β = 3). Distributions were scaled to the range of observed sizes found in this study (minimum and maximum shell widths). For each sampling simulation, six random clams were simulated from each distribution, and toxin concentration was estimated using the empirical relationships with size calculated above. The total toxin content (µg) and concentration (µg STX-eq 100 g^−1^) from each simulation were calculated, and the percentage contribution of each individual clam to the overall toxin concentration of the homogenized sample was derived.

Statistics were analyzed and figures created using the R programming language (version 4.3.1) and the *tidyverse* ecosystem [48,49], where the α level for parametric statistics was 0.01. Graphics were made using the *ggplot2* package [50].

## Figures and Tables

**Figure 1 toxins-16-00464-f001:**
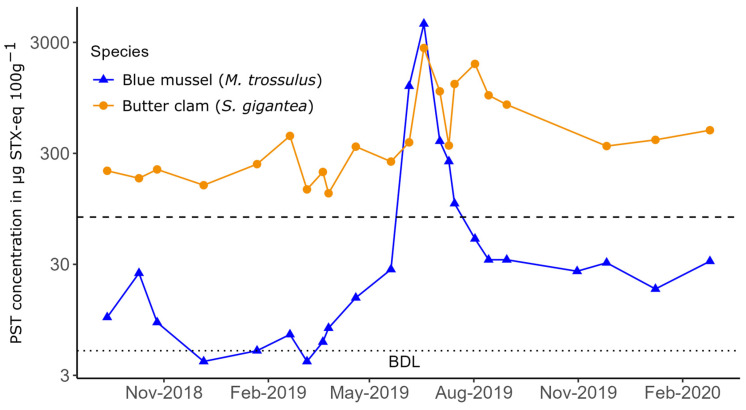
Concentrations of PSTs in butter clams and blue mussels measured at one site near Juneau, Alaska by the CCTHITA. The dashed line at 80 µg STX-eq 100g^−1^ is the US FDA limit for human consumption, and the dotted line at 5 µg STX-eq 100g^−1^ is the detection limit of the assay. Note the log scale on the y-axis.

**Figure 2 toxins-16-00464-f002:**
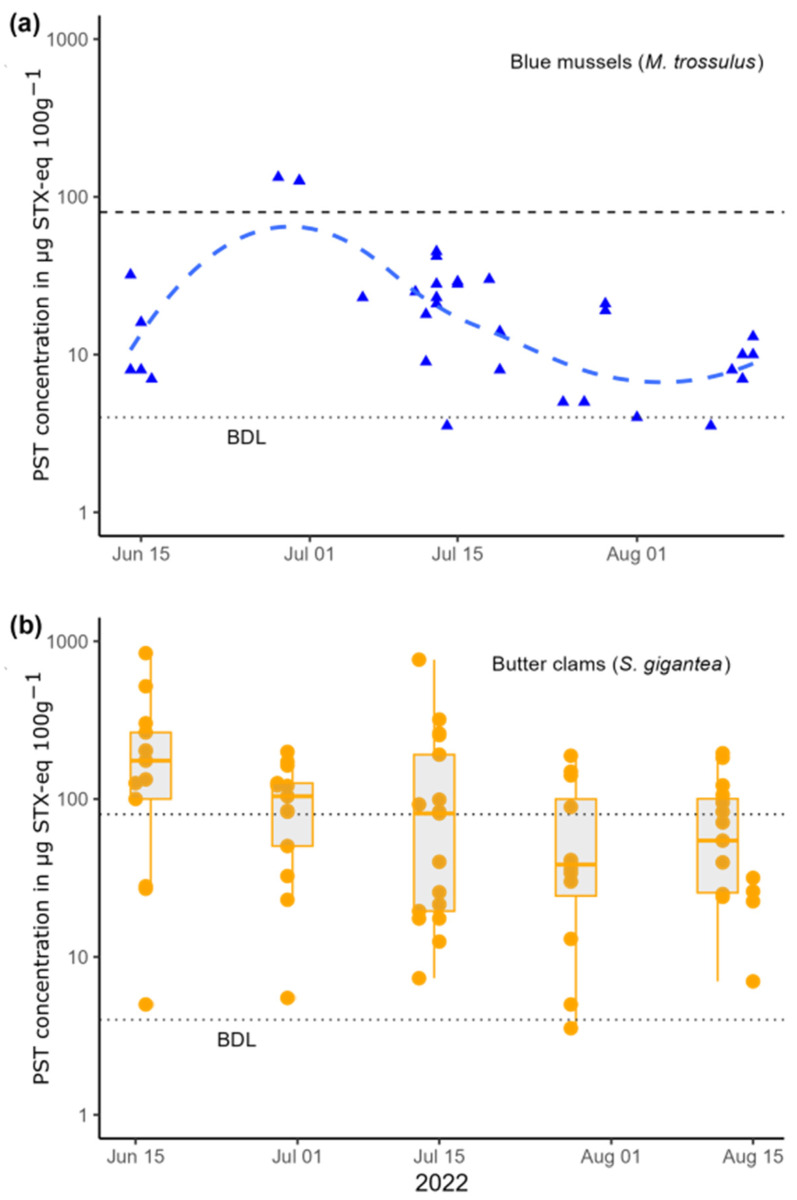
(**a**) Concentrations of PSP toxins in blue mussels measured from SEATOR partners in 2022 (blue points, data from SEATOR) and trend line calculated using locally estimated scatterplot smoothing (LOESS, dashed line). (**b**) Concentrations of PSP toxins in butter clams measured in this study (orange points), boxplots are also shown for distributions for each sampling period (5)—note the log scale on both y-axes. The dashed line at 80 µg STX-eq 100 g^−1^ is the US FDA limit for human consumption, and the dotted line at 5 µg STX-eq 100 g^−1^ is the detection limit of the assay.

**Figure 3 toxins-16-00464-f003:**
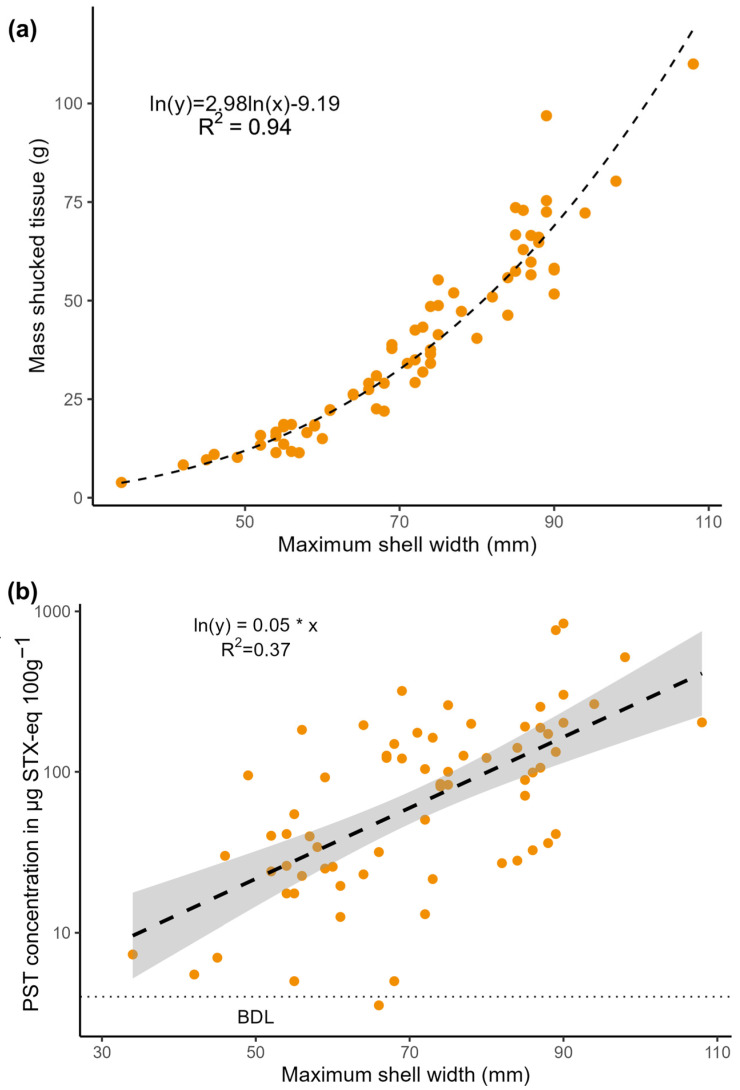
(**a**) Relationship between width and mass for individual butter clams sampled in this study. (**b**) Concentrations of PSTs in butter clams (whole tissue, wet weight) in relation to maximum shell width, shaded region represents 95% confidence band for the regression. Note log scale on y-axis.

**Figure 4 toxins-16-00464-f004:**
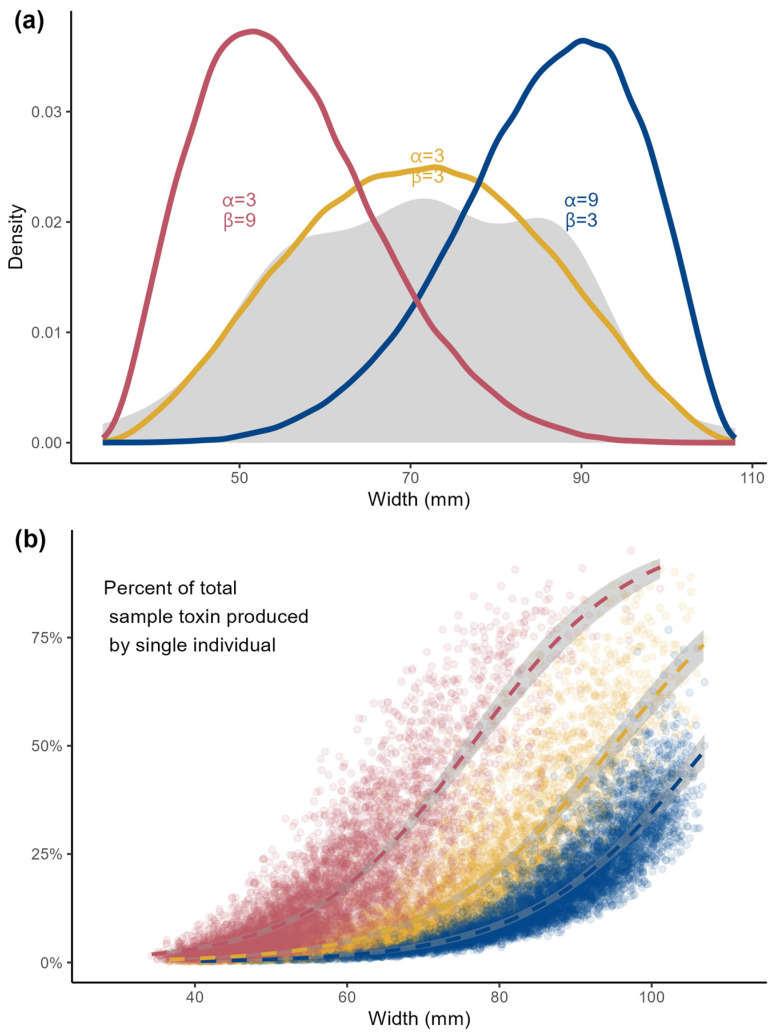
(**a**) Density functions of simulated β distributions (colored lines) and observed density function for butter clam shell width (gray shaded region) for this study, (**b**) percent of total toxicity of a 6-clam homogenate contributed by a single clam as a function of width (n = 1000 simulations for each distribution). Smooth curves are binomial logistic regression models for each distribution type, and shaded regions represent confidence intervals.

## Data Availability

The raw data supporting the conclusions of this article will be made available by the authors on request.

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
