# Peer review of "Clam Size Explains Some Variability in Paralytic Shellfish Toxin Concentrations in Butter Clams (Saxidomus gigantea) in Southeast Alaska"

_toxins, 2024, doi:10.3390/toxins16110464_

Round 1
Reviewer 1 Report (Previous Reviewer 2)
Comments and Suggestions for Authors
Thank you for adding information regarding the possible influence of different STX congeners.
Some minor comments:
Line 58,59. This makes it sound like the DOH is still using the mouse bioassay which highly unlikely. The ref is 2009.
Line 62: This ref is 2009 so there will be more recent info on the Acute reference doses (ARfD) etc
Line 78: Mistake with "in in"
Line 83: Mistake with "individualsvand"
The figures on page 5 are messed up. Figure 2 last line of the caption "(5) - note the log scale" and then it looks like the next figure overlays part of the caption.
Figure 3. The caption refers to b) but I can see only 1 graph.
Line 173: You talk about toxicity. "the difference in toxicity" "the RBA and actual toxicity" The difference is STX eq value
Line 183, 184: I think this sentence needs to be rewritten, it is not clear.
Line 186: Spelling mistake "equivalncy"
Line 186: I think toxicity equivalency factors need to be define and their use explained. This would perhaps be best in the introduction. You could talk about STX congeners have different toxicities, this is corrected for analytically by use of TEFs which .... The binding efficiencies for some congeners, espectially .. do not correlate with toxicity.
Comments on the Quality of English LanguageSome minor spelling mistakes to be corrected
Author Response
Thank you for adding information regarding the possible influence of different STX congeners.
Some minor comments:
Line 58,59. This makes it sound like the DOH is still using the mouse bioassay which highly unlikely. The ref is 2009.
We have provided an updated reference to indicate that the DOH is still in fact using the mouse bioassay (MBA).
Line 62: This ref is 2009 so there will be more recent info on the Acute reference doses (ARfD) etc
Lethal dose calculations for PSP toxins in humans are difficult to determine, and more recent reporting does not necessarily indicate higher accuracy. The value cited here (0.6 µg/kg bw) is similar to values presented in more recent studies (e.g. Arnich & Thebault, 2018), and a thorough discussion of variability around these estimates is beyond the scope of this paper, since this statement is only meant to indicate the potential for lethal doses of toxins from individual clams.
Line 78: Mistake with "in in"
This change has been made.
Line 83: Mistake with "individualsvand"
This change has been made.
The figures on page 5 are messed up. Figure 2 last line of the caption "(5) - note the log scale" and then it looks like the next figure overlays part of the caption.
There was a mistake with the PFD rendering upon upload to Toxins, this will be remedied with the revised resubmission.
Figure 3. The caption refers to b) but I can see only 1 graph.
There was a mistake with the PFD rendering upon upload to Toxins, this will be remedied with the revised resubmission.
Line 173: You talk about toxicity. "the difference in toxicity" "the RBA and actual toxicity" The difference is STX eq value
Since the receptor binding assay is dependent on a functional response of a target tissue (pig brain homogenate) and calibrated to the functional response of saxitoxin, the resulting data are expressed as STX-eqivalent units. However this is a determination of composite toxicity, and the term “toxicity” is used extensively in literature for this assay (van Dolah et al. 2012, Ruberu et al, 2018, Turner et al., 2018, many others). We have added a statement to reflect that by actual toxicity we’re referring to observed toxicity in animal models (e.g. mouse bioassay).
Line 183, 184: I think this sentence needs to be rewritten, it is not clear.
We have edited the sentence to now read:
Although there may be discrepancies between binding affinities and observed toxicities for some congeners, the agreement of RBA and HPLC across a wide range of total toxin concentrations suggests that results obtained from the RBA are not severely overestimating total toxicity as compared to HPLC.
Line 186: Spelling mistake "equivalncy"
We have edited this sentence for clarity.
Line 186: I think toxicity equivalency factors need to be define and their use explained. This would perhaps be best in the introduction. You could talk about STX congeners have different toxicities, this is corrected for analytically by use of TEFs which .... The binding efficiencies for some congeners, espectially .. do not correlate with toxicity.
We added an explanation of STX-equivalents in the introduction. We feel that a definition of TEFs broadly should be available from many sources to readers of the journal if they’re unfamiliar with the concept. The discussion of binding efficiencies as related to toxicities is done in the discussion where it follows naturally the description of the methodology and results. We feel that a more thorough discussion of the intricacies of brining affinities and toxicity is outside the scope of this communication.
Reviewer 2 Report (Previous Reviewer 3)
Comments and Suggestions for Authors
General comments:
It is an interesting paper. The authors made the changes suggested in the first revision. This work can be published after making some modifications. I have some minor comments that can help improve the manuscript. Here are some suggestions in detail:
Specific commentaries:
Abstract
Well written and clear.
Line 13. What species of Alexandrium? Not all are toxic
Line2 16-17. What is a period of relatively stable PST concentrations and HAB activity?
Introduction
Line 83. Replace “individualsvand” by “individuals and”.
Discussion
Line 138. What species of Alexandrium? Not all are toxic
Line 146. Delete “significantly”. Is redundant.
Line 147. Include “(p<0.05)” after “statistically significant”.
Line 186. Replace “equivalncy" by “equivalency”
Material and methods
Line 231. Please explain “negative tides”? Did you refer to low tide?
Figures
Figure 2. Why do you use different graphs (scatterplot and boxplot) to represent the same variable?
Author Response
General comments:
It is an interesting paper. The authors made the changes suggested in the first revision. This work can be published after making some modifications. I have some minor comments that can help improve the manuscript. Here are some suggestions in detail:
Specific commentaries:
Abstract
Well written and clear.
Line 13. What species of Alexandrium? Not all are toxic
This is likely Alexandrium catenella based on previous studies, however this nor the SEATOR monitoring program identifies Alexandrium cells to species level. The species distinction is not necessary for this paper, and the presence of PSP toxins affirms that the species of Alexandrium seen in SE Alaska is indeed toxigenic. We added some text below to respond to the same concern from this reviewer.
Line2 16-17. What is a period of relatively stable PST concentrations and HAB activity?
We have removed the phrase HAB activity, the stable PST concentrations were described in the paper using results from monitored species (blue mussels), which are used as an sentinel species for their ability to uptake and depurate toxins rapidly.
Introduction
Line 83. Replace “individualsvand” by “individuals and”.
This has been done.
Discussion
Line 138. What species of Alexandrium? Not all are toxic
We have added that this is likely Alexandrium catenella based on previous studies.
Line 146. Delete “significantly”. Is redundant.
This change has been made.
Line 147. Include “(p<0.05)” after “statistically significant”.
We have made this edit.
Line 186. Replace “equivalncy" by “equivalency”
We have made this change as suggested by another reviewer.
Material and methods
Line 231. Please explain “negative tides”? Did you refer to low tide?
We have added the word “low” in the parenthetical, but negative tides are explained in the immediately following its use – butter clams are not available during all low tides. Often during quarter and gibbous moon phases the low-low tides (Juneau has mixed semi-diurnal tides) are ~2m and butter clams are not accessible to be hand collected. Only during negative tides (tides <0m) can clams be reliably collected.
Figures
Figure 2. Why do you use different graphs (scatterplot and boxplot) to represent the same variable?
Data collected in Figure 2a was collected as part of SEATOR monitoring efforts and represents blue mussels, collected at varying time points. Data collected in Figure 2b is specific to this study and was collected over 5 discrete sampling efforts, shown by the boxplots. The data in Figure 2b are also presented in scatterplot form (points), but the boxplots are shown to illustrate statistically the analysis which is later done to compare concentrations through time.
Reviewer 3 Report (Previous Reviewer 5)
Comments and Suggestions for Authors
Dear Editor Greetings,
I have read in detail the new version of the manuscript entitled “Clam size explains some variability in paralytic shellfish toxin concentrations in butter clams (Saxidomus gigantea) in Southeast Alaska” (Type of manuscript: Communication).
The authors in this new version have answered questions and corrected details in the text. However, unresolved experimental issues remain that need to be included even though the manuscript type corresponds to Communication. Additionally, the discussion explores aspects of toxicity that according to the method used could only be established as assumptions without the data provided clearly allowing to resolve the questions.
I suggest giving a new revision to the materials and methods section, to be considered in a new admission.
Comments on the Quality of English Language
Moderate editing of English language required.
Author Response
Dear Editor Greetings,
I have read in detail the new version of the manuscript entitled “Clam size explains some variability in paralytic shellfish toxin concentrations in butter clams (Saxidomus gigantea) in Southeast Alaska” (Type of manuscript: Communication).
The authors in this new version have answered questions and corrected details in the text. However, unresolved experimental issues remain that need to be included even though the manuscript type corresponds to Communication. Additionally, the discussion explores aspects of toxicity that according to the method used could only be established as assumptions without the data provided clearly allowing to resolve the questions.
I suggest giving a new revision to the materials and methods section, to be considered in a new admission.
The discussion of whether the methodology (RBA) could be driving observed differences in toxicity was requested by several reviewers from the first round of revision. It seems like the reviewer is asking for data to compare these samples between HPLC and RBA to assess this hypothesis directly, but we provide support in the discussion that we don’t believe this to be the case – and furthermore a thorough comparison of the methods is outside the scope of this paper.
Reviewer 4 Report (New Reviewer)
Comments and Suggestions for Authors
The paper describes the concentrations of PSP were analyzed in association with morphometrics (shell width and shucked mass) for butter clams (Saxidomus gigantea) in Southeast Alaska. The authors found that larger clams had higher concentrations of PST during periods of low harmful algal bloom activity. These results have the potential to impact harvest strategies and monitoring program protocols. I believe that the results are very useful in the field to marine toxins. Thus, I recommend the publication in Toxins. Aside from the recommendation, the authors should clear the following points before publication.
1) I recommend the authors should include the section “Conclusions” after the Discussion session to clarify the author’s conclusions.
2) In line 78, “in in “should be “in”. The authors thoroughly check the whole manuscript again.
3) I think that Figure 3b) is missing.
4) In the legend of Figure 2, after “note the log scale on both y-“ some words should be missing. Please check.
Author Response
The paper describes the concentrations of PSP were analyzed in association with morphometrics (shell width and shucked mass) for butter clams (Saxidomus gigantea) in Southeast Alaska. The authors found that larger clams had higher concentrations of PST during periods of low harmful algal bloom activity. These results have the potential to impact harvest strategies and monitoring program protocols. I believe that the results are very useful in the field to marine toxins. Thus, I recommend the publication in Toxins. Aside from the recommendation, the authors should clear the following points before publication.
- I recommend the authors should include the section “Conclusions” after the Discussion session to clarify the author’s conclusions.
We note that conclusions are occasionally present in Communications in Toxins, and therefore have added the following conclusion:
We found evidence that PSP toxin concentration was positively correlated with butter clam size, and that large butter clams can contribute a disproportionate amount of toxin to homogenate samples collected for routine monitoring. If this relationship holds under other conditions and in other regions, these results could have implications for harvesters and HAB monitoring programs.
- In line 78, “in in “should be “in”. The authors thoroughly check the whole manuscript again.
We have made this edit based on another reviewer’s comments.
- I think that Figure 3b) is missing.
This was a mistake with the conversion to PDF on the submission site, it will be rectified with the resubmission.
- In the legend of Figure 2, after “note the log scale on both y-“ some words should be missing. Please check.
This was a mistake with the conversion to PDF, it will be rectified with the resubmission.
Reviewer 5 Report (New Reviewer)
Comments and Suggestions for Authors
Harmful algal blooms (HABs) pose a significant threat to human health and aquaculture in Southeast Alaska. Our local organization's monitoring programs have greatly improved our understanding of the environmental drivers and toxicokinetics of shellfish toxins in the region. However, there remains a significant variability in shellfish toxins in some species, which cannot be easily explained by seasonal bloom dynamics. To investigate potential sources of this variability in PST concentrations from this subsistence species, our study assessed individual concentrations of PSTs across a size gradient of butter clams during a period of relatively stable PST concentrations and HAB activity. The results revealed that larger clams had higher concentrations of paralytic shellfish toxins during periods of low harmful algal bloom activity. This finding has the potential to significantly impact aquaculture in Southeast Alaska, making our work particularly relevant to the HAB researchers and aquaculture farmers in the region. 1 How to sample and how many clams were used for PST analysis. In the Abstract section, 6 clams were selected to detect the PST, while 6-12 samples individuals were taken, totaling 70 clams across the study period. It is confused. 2 Regarding clam size, how do measure the size?
Author Response
Reviewer 5
Harmful algal blooms (HABs) pose a significant threat to human health and aquaculture in Southeast Alaska. Our local organization's monitoring programs have greatly improved our understanding of the environmental drivers and toxicokinetics of shellfish toxins in the region. However, there remains a significant variability in shellfish toxins in some species, which cannot be easily explained by seasonal bloom dynamics. To investigate potential sources of this variability in PST concentrations from this subsistence species, our study assessed individual concentrations of PSTs across a size gradient of butter clams during a period of relatively stable PST concentrations and HAB activity. The results revealed that larger clams had higher concentrations of paralytic shellfish toxins during periods of low harmful algal bloom activity. This finding has the potential to significantly impact aquaculture in Southeast Alaska, making our work particularly relevant to the HAB researchers and aquaculture farmers in the region.
- How to sample and how many clams were used for PST analysis. In the Abstract section, 6 clams were selected to detect the PST, while 6-12 samples individuals were taken, totaling 70 clams across the study period. It is confused.
This information is presented in the methods. The sentence in the methods may have been confusing, so we have edited the sentence in the materials and methods to now read:
…with 6-12 individuals were taken during each sampling effort totaling 70 clams across the study period.
- Regarding clam size, how do measure the size?
From the materials and methods:
Clams were placed in a cooler and transported back to the lab where they were measured for width across the widest axis of their shells using calipers to the nearest mm.
Reviewer 6 Report (New Reviewer)
Comments and Suggestions for Authors
This article presents a positive relationship between butter clam size and toxin concentration. The toxin concentration was measured by RBA method and the clam size was expressed as the clam width. The study period is from Jun. 15 to Aug. 15. In general, PSP in shellfish was known to be detoxicated in 1-2 weeks. However, the authors suggests that the case in butter clam is different, retaining for long time. Based on this phenomenon, a larger clam can store more amount toxins from toxic phytoplanktons. However, the suggestion that large-sized clams have higher toxin concentration is not persuasible.
This article is short of sufficient data and a critical point is to miss a important data (Figure 3b). The correlation value (R2) for size to toxin concentration is so low that the trend can't be accepted.
Comments on the Quality of English LanguageEnglish in this article is good, but scientifically not expressed that the readers are hard to understand.
And the caption of Figure 2 was cut, and Figure 3b was not presented.
Author Response
This article presents a positive relationship between butter clam size and toxin concentration. The toxin concentration was measured by RBA method and the clam size was expressed as the clam width. The study period is from Jun. 15 to Aug. 15. In general, PSP in shellfish was known to be detoxicated in 1-2 weeks. However, the authors suggests that the case in butter clam is different, retaining for long time. Based on this phenomenon, a larger clam can store more amount toxins from toxic phytoplanktons. However, the suggestion that large-sized clams have higher toxin concentration is not persuasible.
The retention of PSP toxins in butter clams is well described in literature, this is not a claim that is made by this paper.
This article is short of sufficient data and a critical point is to miss a important data (Figure 3b). The correlation value (R2) for size to toxin concentration is so low that the trend can't be accepted.
We are not suggesting that size is the only factor that can explain variation in PSP toxin concentrations, and we spend a great deal of time in the manuscript suggesting other sources of variability. The relationship with size, while not the only driver of variability, is an important one to assess because it is a) highly influential on the monitoring protocols in use (generating a 6 clam homogenate) and b) immediately apparent to harvesters. If there are predictable relationships with size, these dynamics should be assessed in more detail and this work is a preliminary attempt at that research.
This study was based on field-collections of wild organisms and is presented as preliminary evidence, we don’t feel that this R2 value should be disqualifying. Indeed, the statistical analysis used indicates that this relationship is highly significant.
Round 2
Reviewer 3 Report (Previous Reviewer 5)
Comments and Suggestions for Authors
The article is very interesting and raises the option of toxic variability among individuals who present a prevalence of PST.
I suggest that the authors make the proposed changes and clearly discuss the following comments.
Comments:
Line 16-21: I suggest rethinking this paragraph, since it contributes negatively to the data included in the article. It is obvious that toxin prevalence data are different from assimilation processes, a phase in which, depending on the physiology of the organism, it could involve biotransformation.
Line 63: Relate this important information in a way that is more linked to the ARfD. To do this, rewrite these lines.
Line 75: What does high concentrations mean? If the value is very significant, it would be very prudent to include it and relate it to the density of microalgae.
Figure 1: Please be more precise in the figure legend. The relevant data that you intend to highlight are included in the text.
Line 86-91: This paragraph is very guiding. However, the reference included is not enough to argue, I suggest you include the ability of some species of clam to assimilate high concentrations of toxins without physiological limitations.
Line 99-102: Please be more precise in your hypothesis.
Figure 2: Please harmonize the Y axis in both figures (a and b).
Figure 4: Please check the quality of the figures included.
Line 143: It seems to me that the most appropriate word is “prevalence.”
Line 247-249: This paragraph partially supports the weakness of the discussion in the text, since it does not adequately address the physiology of hydrobiological organisms to assimilate, retain, distribute, biotransform and eliminate toxins. Even when blooms are abundant, living organisms do not feed 24 hours a day. Some reject dinoflagellates as pseudofeces.
Line 260-264: Based on the previous comment, he raises his conclusion in a not so drastic way, since there are parameters that he has not evaluated.
Line 274: At what temperature?
Line 278-281: Does this process have any reference to support it?
Line 283-284: Delete this line and keep the reference next to (36,37).
Line 265: Include in this section the biosafety and security regarding the disposal of biological and radioactive waste.
Comments on the Quality of English LanguageModerate editing of English language required.
Author Response
See response attached

Reviewer 5 Report (New Reviewer)
Comments and Suggestions for Authors
Thanks for the authors' response. My concern was solved.
Author Response
See response attached
Reviewer 6 Report (New Reviewer)
Comments and Suggestions for Authors
I think your current reports were listed well.
Author Response
See response attached
This manuscript is a resubmission of an earlier submission. The following is a list of the peer review reports and author responses from that submission.
Round 1
Reviewer 1 Report
Comments and Suggestions for Authors
This manuscript "Clam size explains some variability in paralytic shellfish toxin concen- 2
trations in butter clams (Saxidomus gigantea) in Southeast Alaska" has been submitted as a Communication to the Journal Toxins.
| The authors concluded that larger clams had higher concentrations of par- | 27 |
|
alytic shellfish toxins during periods of low harmful algal bloom activity. Since they have studied samples of 1 individual clam, and selected 6 isolated invividuals for the expriments it seems of relevant poor statistical significance. The information is descriptive, neither any clear conclusion of ecological, physiological or biochemical new information is provided. Thus, it is sugested to the authors to complete their studies to increase the relevance of the information presented, before submitting the manuscript. Moreover, since the study was conducted in the absence of HAB with low harmful algal bloom activity this Journal does not seem as the more approriate journal option. It should be considered by the authors. |
Reviewer 2 Report
Comments and Suggestions for Authors
I have reviewed the paper "Clam size explains some variability in paralytic shellfish toxin concentrations in butter clams (Saxidomus gigantea) in Southeast Alaska" and found it to be well-written and easy to read.
However, I have one major concern with the study. The analysis method used was the RBA and it is known that the binding affinities of the different PST analogues (NeoSTX=1.45, dcSTX=10.26 and GTX5=32.5) are not the same as the toxicity equivalence factors (TEFs) used by EFSA (NeoSTX=2, dcSTX=0.5 and GTX5=0.1). The EFSA TEFs have been calculated to account for the different toxicities of analogues and allow the expression of concentration as STX (eq), the regulatory limit is expressed as STX (eq). Do you have any data (yours or from the literature) about how the percentage of the different analogues is affected by shellfish size? From your data it is possible that it is not the total PSTs that is increasing with clam size but that there is an increasing percentage of GTX5 or dcSTX. Since these two analogues have very high binding affinities in the RBA the total PSTs of a sample containing them will be greatly overestimated.
In the introduction you say "there are more than two dozen chemical congeners" when in fact there are >50 analogues now known, I think you need a more recent reference?
My other comments are only minor:
Line 48 does not make sense.
Line 70 should be rewritten "The large blooms observed in an extremely high toxin concentrations..."
The references contained some mistakes. Species names need to be italicized throughout, some journals are abbreviated and some not and some abbreviations had full stops and some not.
Comments on the Quality of English Language
The English is good
Reviewer 3 Report
Comments and Suggestions for Authors
Reviewer 4 Report
Comments and Suggestions for Authors
General
Paralytic shellfish toxins, produced by certain phytoplankton species, are one of the major problems affecting aquaculture and public health around the world. There have been substantial advancements in the refinement of the official detection methods as well as understanding species-specific toxin accumulation.
The authors of this study aimed at suggesting that clam size could potentially reflect PST concentration; however, the very small sample size (n=70) for the scope of this study plus the lack of a bloom made it difficult to demonstrate the hypothesis. They mostly used models to explain or prove their hypothesis, but despite their model showing a significant correlation between clam size and toxin concentration, the coefficient of determination (r2=0.37) was low to also support the hypothesis. These results cannot be taken lightly given the high risk in public health. A much larger sample size is required including a wide toxin range of naturally exposed clams to obtain better and reliable models.
Particular
Line 58. Mention the exact official method used by the Washington Department of Health (which compares to the one used in the manuscript), with corresponding reference.
Line 61. Include full units, as 80 µg STX-eq 100 g-1 (or STX-2HCl eq, depending on the reporting in Alaska or USA in general). Please correct the units elsewhere in the manuscript.
Line 71. Include that near lethal dose for humans.
Figure 1. Explain the two interrupted/dotted lines, what are they showing in the figure? If one of them highlights the regulatory limit, specify, but if not, please include it. The legend of the y axis needs to be corrected, perhaps the authors meant µg STX-eq 100 g-1. The concentrations mentioned in the figure legend also need to have the proper and complete units as mentioned previously.
Line 92. I believe the authors meant 80 µg STX-eq 100 g-1.
Figure 2. Similar to Fig. 1, explain the dotted line.
Figure 3. As previously mentioned, the correct units (and order) are µg STX-eq 100 g-1.
Line 144. Alexandrium (genus) must be in italics.
Lines 160-162. This statement seems biased and questionable about the monitoring of PST. Given that sampling for toxin testing should be random, at the very least, the organisms collected may have plate size. Is this a reference for this statement?
Lines 189-191. How significant (p value) is this when including a low coefficient of determination? What are the major limitations of the model including both parameters? The authors aimed at suggesting that clam size could potentially reflect PST concentration; however, the very small sample size (n=70) for the scope of this study plus the lack of a bloom made it difficult to demonstrate the hypothesis.
Comments on the Quality of English LanguageMinor erros found.
Reviewer 5 Report
Comments and Suggestions for Authors
Dear Editor Greetings.
I report that I have read the Communication entitled: "Clam size explains some variability in paralytic shellfish toxin concentrations in butter clams (Saxidomus gigantea) in Southeast Alaska".
The authors hypothesize: "Concentrations of PSTs in individual clams are related to morphometrics (mass, width) in butter clams collected in Southeast Alaska, and examine broader implications of these results using bootstrapped sampling simulations".
The article is interesting, however, there are aspects that the authors need to clarify.
The variability in the toxicity related to PST is a common event in different filtering species, which brings into play the representativeness of the sample collection to be analyzed, since each individual can show remarkable variability in toxicity. There is also an aspect to clarify, toxicity is the total presentation of the analogues present in a sample, but the detoxification process can be dependent on the distribution of the accumulated analogues in the bivalve tissues (visceral and non-visceral), which adds to the bioconversion of each analogue in each process. It is important to note that the clearance rate is a polarity-dependent process of the analogs. Therefore, the non-correlation in lower toxicity could be related to the bioconversion of analogs that enhance total toxicity over time.
Another important aspect is to establish the profile of toxins contributed by the dinoflagellate species associated with PST in the study area.
Additionally, consider that the species evaluated is from a sandy bottom habitat and therefore, the incorporation of detritus and toxic material from other species may eventually contribute to the toxicity of the species.
It is important that the authors discuss these points.
I also include the following comments:
Line 41-43: Please rewrite this section, the idea put forward is difficult to understand.
Line 48: Correct the punctuation on this line.
Line 50: What is the percentage value of the threshold you have exceeded (include the regulatory limit).
Line 61: FDA considers equivalence at the regulatory limit?
Figure 1: Review the toxicity of the figure (Y-axis and legend). Indicate what is BDL and the meaning of the dotted lines.
Line 82. Include a reference to this important phrase.
Lie 84: Include the reference values determined, for clarity of the toxicities detected.
Line 90: Review the expression of toxicities throughout the section. It is usual to use μg STX eq 100 g-1.
Line 91-93: Please revise this paragraph and write it clearly.
Figure 2: Review the toxicity of the figure (Y-axis). Indicate the meaning of the dotted lines.
Figure 3b: Review the toxicity of the figure (Y-axis).
Line 93: Please be more precise in the limit of detection and quantification of the assay used and also explain how the matrix effect was evaluated.
Comments on the Quality of English LanguageModerate editing of English language required